# AniScan Using Extracellular Cyclic AMP-Dependent Protein Kinase A as a Serum Biomarker Assay for the Diagnosis of Malignant Tumors in Dogs

**DOI:** 10.3390/s20154075

**Published:** 2020-07-22

**Authors:** Ji-Eun Lee, Woo-Jin Song, Hunjoo Lee, Byung-Gak Kim, Taeho Kim, Changsun Lee, Bonghwan Jang, Hwa-Young Youn, Ul-Soo Choi, Dong-Ha Bhang

**Affiliations:** 1Department of Veterinary Clinical Pathology and Bio-Safety Research Institute, College of Veterinary Medicine, Chonbuk National University, Jeonju 54896, Korea; clinpath_ez@naver.com; 2Department of Veterinary Internal Medicine and Research Institute of Veterinary Science, College of Veterinary Medicine, Jeju National University, Jeju 63243, Korea; ssong@jejunu.ac.kr; 3Biattic Inc., Anyang 14059, Korea; leehunz@biattic.com (H.L.); bk@biattic.com (B.-G.K.); 4AID Animal Hospital, Seoul 06032, Korea; angelmeo@gmail.com (T.K.); wasser37@naver.com (C.L.); 5Goodmorning Pet Hospital, Seongnam 13613, Korea; drummer75@hanmail.net; 6Laboratory of Veterinary Internal Medicine, Department of Veterinary Clinical Science, College of Veterinary Medicine, Seoul National University, Seoul 08826, Korea; hyyoun@snu.ac.kr; 7Department of Molecular and Cellular Biology, Samsung Biomedical Research Institute, Sungkyunkwan University School of Medicine, Suwon 16419, Korea

**Keywords:** extracellular c-AMP-dependent protein kinase A, biomarker, dog, cancer, tumor

## Abstract

The early detection of tumors improves chances of decreased morbidity and prolonged survival. Serum biomarkers are convenient to use and have several advantages over other approaches, such as accuracy and straightforward protocols. Reliable biomarkers from easily accessible sources are warranted for the development of cost-effective assays for routine screening, particularly in veterinary medicine. Extracellular c-AMP-dependent protein kinase A (ECPKA) is a cytosolic leakage enzyme. The diagnostic accuracy of detecting autoantibodies against ECPKA was found to be higher than that of ECPKA activity from enzymatic assays, which use a complicated method. Here, we investigated the diagnostic significance of measuring serum ECPKA autoantibody levels using an in-house kit (AniScan cancer detection kit; Biattic, Anyang, Korea). We used sera from 550 dogs, including healthy dogs and those with malignant and benign tumors. Serum ECPKA and immunoglobulin G were determined using the AniScan cancer detection kit. ECPKA autoantibody levels were significantly higher (*p* < 0.01) in malignant tumors than in benign tumors, non-tumor diseases, and healthy controls. On the basis of sensitivity and specificity values, AniScan ECPKA is a rapid and easy-to-use assay that can be applied to screen malignant tumors from benign tumors or other diseases in dogs.

## 1. Introduction

The early detection of tumors can decrease morbidity, prolong survival, and provide better opportunities and options for treatment [1,2]. Extensive efforts to identify biomarkers for the early detection of tumors have been successful, and several biomarkers are used to diagnose tumors in subclinical stages in human medicine [3], as well as in veterinary medicine [4,5,6]. These biomarkers might involve DNA, mRNA, proteins, metabolites, or processes associated with tumor development [6]. These biomarkers include tumor antigens such as alpha-fetoprotein (AFP); biochemical enzymes (alkaline phosphatase, lactate dehydrogenase), acute-phase proteins (alpha 1-acid glycoprotein, C-reactive protein, serum amyloid A, and haptoglobin), and leakage enzymes such as serum thymidine kinase 1 (TK-1), high-mobility group B1 proteins (HMGB1), or extracellular c-AMP-dependent protein kinase A (ECPKA) [4,5].

For the diagnosis of solid tumors, fine-needle aspiration cytology is a non-invasive, inexpensive, and rapid tool [7], and histology is considered the gold standard. However, the accuracy of cytology is limited by the quality of the sample or by sample representability, especially when the mass is deep in the abdomen, is extremely large, or is accompanied by inflammatory processes, which obfuscates the discrimination between inflammation and neoplasm. Anesthesia or surgical procedures are needed to execute histological examination. However, the use of biomarkers detectable in serum will be convenient for veterinarians and patients as long as the diagnosis is accurate.

Cyclic AMP (cAMP)-dependent protein kinase A (PKA) is a serine/threonine protein kinase composed of two catalytic (C) and two regulatory (R) subunits. There are two types of PKAs, type I (PKA-I) and type II (PKA-II), depending on two different regular subunits, RI and RII, respectively, with common catalytic subunits. PKA mediates the cAMP-related pathway and is involved in cell proliferation, differentiation, metabolism, and apoptosis in mammalian cells [8,9,10,11]. In normal cells, PKA exists as an intracellular enzyme. However, under malignant conditions, the ratio of PKA-I to PKA-II changes, and its catalytic subunits are secreted into the extracellular environment [8]. This is referred to as extracellular cAMP-dependent protein kinase A (ECPKA). Evidence shows that the activity of ECPKA increases in various malignant tumors in human medicine [8,9,10,11], and a report demonstrated that ECPKA activity decreased after surgical removal in human melanoma patients [12]. However, although it is suggested that the ECPKA level also increases in malignant tumors in mammals other than humans, few studies regarding ECPKA are available in the veterinary literature [13,14].

As cancer cells proliferate, ECPKA is secreted into the extracellular compartment, followed by the induction of autoantibodies against ECPKA. These autoantibodies are amplified and produced in relatively large amounts by the immune system, making their detection easy [15]. The enzyme immunoassay (EIA), which measures the immunoglobulin G (IgG) autoantibody against ECPKA, was developed and applied to the diagnosis of malignant tumors. High anti-ECPKA autoantibody titers were found in the sera of malignant tumor patients, whereas low or negative titers were reported in the control group. The diagnostic accuracy of detecting autoantibodies against ECPKA was higher than that of measuring ECPKA activity by the enzymatic assay [8]. Therefore, detecting the level of the anti-ECPKA autoantibody in serum might be more advantageous than the ECPKA enzymatic assay, which uses a more complicated method [16].

The aims of this study were to evaluate the diagnostic significance of measuring serum ECPKA autoantibodies using an in-house kit (AniScan cancer detection kit) and to determine whether ECPKA can fulfill the role of malignant tumor detection in dogs.

## 2. Materials and Methods

### 2.1. Animals and Samples

Sera were collected from privately owned dogs that visited the Chonbuk Animal Medical Center and several local animal hospitals in the Republic of Korea from May 2018 to June 2019. The dogs included in the control group were considered clinically healthy based on no remarkable findings in complete blood count, serum biochemistry, or thoracic and abdominal radiographic examination. Based on the history taken, dogs that were confirmed to have been treated with any corticosteroids were omitted from the study population.

### 2.2. Diagnosis of Malignant and Benign Tumors

The diagnosis of malignant and benign tumors was performed by cytology and/or histology. For the cases where histopathology data were available, tissue specimens were obtained from surgical excision or biopsy of the tumors, which were treated with neutral buffered 10% formalin and fixed. The samples were sent to Korea Vet Lab (Seongnam, Korea) to conduct histopathological evaluations and reach a definitive diagnosis. Cytological specimens were obtained by capillary aspiration or ultrasound-guided fine-needle aspiration using 23 G needles and 3–5 mL syringes. The smears were produce by squash or line smear methods according to routine cytological slide preparation procedures, air-dried, fixed with methanol, and stained with Diff-Quik®. All cytological specimens were reviewed by a clinical pathologist, and the diagnosis was made by cytology alone and/or by histopathology. For lymphoma, cytological smears were sent to Korea Vet Lab to perform PCR for antigen receptor gene rearrangement (PARR) for the identification of cell type and definite diagnosis.

### 2.3. Assessment of Serum ECPKA Autoantibody

The serum ECPKA autoantibody was determined using the AniScan cancer detection kit (Biattic Inc., Anyang, Korea) based on a lateral flow immunochromatographic assay (LFIA). In brief, 10 µL of serum was diluted 1:500 with 5 mL of phosphate-buffered solution (pH 7.0–7.4) and mixed well. 

We dropped 100 µL onto the sample window, which was passed through a nitrocellulose membrane pad. Serum ECPKA autoantibodies conjugated with 50 nM gold nanoparticles were bound by canine PKA antigen (National Center for Biotechnology Information reference sequence: NM_001003032.1) pre-coated on the test line. Then, the result appeared as red lines, and strength of the red color was measured by using computer vision techniques (Aniscaner, Biattic Inc.) that could find the exact location of red lines on the kit image (Figure 1). The system calculates the areas of the red lines, and the values of the areas stand for intensities of the lines. The level of the ECPKA autoantibody is expressed as unit intensity (UI). 

### 2.4. Statistical Analysis

Differences among more than two groups were analyzed by one-way ANOVA, and the *t*-test was used to analyze differences between two groups, all of which were in parametric distribution. All graphs are presented as box and whisker plots. All the data are shown as the median and range obtained in at least three independent experiments. The statistical analyses were performed using SPSS Statistics 25 (IBM Corp., Chicago, IL, USA) and GraphPad Prism version 8 (GraphPad Software, La Jolla, CA, USA). A *p*-value of 0.05 was considered statistically significant. 

## 3. Results

Sera were obtained from 550 dogs: 227 were diagnosed with malignant tumors and 323 were diagnosed as having non-malignant tumors (60 with benign tumors, 79 with non-tumor disease, and 184 healthy dogs). Their signalment data are summarized in Table 1. The median ages were 11, 10, 8, and 5 years in the malignant tumor group, benign tumor group, non-tumor disease group, and healthy group, respectively. The dogs of tumor-bearing groups were significantly older than those in the non-tumor groups (*p* < 0.001, *t*-test). In addition, neutered females (n = 177) and neutered males (n = 211) accounted for a major part of the population (females, n = 96; males, n = 66). The malignant tumors were categorized into carcinoma (n = 124), sarcoma (n = 50), and hematopoietic (n = 49) or neuroendocrine (n = 4) tumor according to cell origin. Carcinomas included malignant mammary gland tumor (n = 59), transitional cell carcinoma (TCC; n = 22), squamous cell carcinoma (SCC; n = 18), adenocarcinoma (n = 15), and others (n = 10). Sarcomas included melanoma (n = 17); malignant mesenchymal cell tumor (n = 16), the cellular origin of which could not be identified solely by cytology; soft tissue sarcoma (STS; n = 12); and hemangiosarcoma (n = 5). Hematopoietic tumors included high-grade lymphoma (n = 30), malignant mast cell tumor (n = 16), and leukemia (n = 3). Four dogs with neuroendocrine tumor had pheochromocytoma. The benign tumors included benign prostatic hyperplasia, lipoma, and benign skin tumor. The disease types and numbers of non-tumor disease are summarized in Table 2.

The ECPKA autoantibody level in sera was analyzed by LFIA. The median value in the malignant tumor group was 135.12 (range: 11.03–1406.10) UI, and in the benign tumor, non-tumor disease, and healthy 70–200 control groups, the values were 49.88 (range: 8.10–270.60) UI, 45.98 (range: 1.00–203.60) UI, and 44.23 (range: 1.80) UI, respectively (Figure 2). The results showed that the ECPKA autoantibody level was significantly higher in dogs with malignant tumors than in the benign tumor, non-tumor disease, and healthy control groups (all, *p* < 0.01). However, the ECPKA autoantibody level between dogs with benign tumors and non-tumor controls did not show statistically significant differences (benign tumor vs. non-tumor disease, *p* = 0.605; benign tumor vs. healthy, *p* = 0.685).

The receiver operating characteristic (ROC) curve was analyzed to evaluate the diagnostic value of ECPKA autoantibody levels and to determine the sensitivity and specificity of the test (Figure 3). The area under the curve (AUC) of the ROC curve was 0.818 (95% confidence interval (CI), 0.777–0.853), and the sensitivity and specificity were 81.08% and 87.02%, respectively, with an accuracy of 84.67%. The diagnostic characteristics are shown in Table 3.

All types of malignant tumors except neuroendocrine cancers showed significant differences in the level of ECPKA autoantibodies when compared to non-tumor controls (all, *p* < 0.001; Figure 4). The median value of the ECPKA autoantibody level in each type was 136.53 (carcinoma, range: 11.00–1192.60, n = 124), 115.35 (sarcoma, range: 18.00–281.10, n = 50), 150.74 (hematopoietic cancer, range: 12.10–1406.10, n = 49), 147.13 (neuroendocrine cancer, range: 76.80–217.70, n = 4), and 45.68 (non-cancer, range: 0.90–270.60, n = 323).

## 4. Discussion

The aim of this study was to demonstrate the increase in AniScan ECPKA antibody levels in dogs bearing malignant tumors compared to non-tumor groups and to evaluate the usefulness of this approach for detecting malignant tumors in dogs. A significantly higher ECPKA antibody level was found in various malignant tumor groups than control groups, as shown in previous studies with regard to canine malignant tumors [13,14]. In human medicine, abundant evidence shows that ECPKA activity or the ECPKA autoantibody level increases in various malignant tumors regardless of the type of tumor or location, including bladder, breast, cervical, colon, esophageal, gastric, liver, lung, ovarian, pancreatic, prostate, renal, renal cell, small bowel, rectal, and adenocystic carcinomas; melanoma; sarcoma and thymoma; liposarcoma; and leiomyosarcoma [8]. This is consistent with the findings of the present study; thus, ECPKA can be used as a universal cancer biomarker and applied to screening malignant tumors from benign tumors or other diseases.

Candidate biomarkers for tumor detection have their own advantages and disadvantages [15,16]. According to a review of serum autoantibody biomarkers for canine cancers [17], certain biomarkers can be adopted in specific types of cancer. For instance, prostate-specific antigen (PSA) or AFP can be only used to detect prostate cancer or liver cancer, respectively. Although the levels of acute-phase proteins such as serum amyloid A, C-reactive protein, or haptoglobin were shown to be elevated in dogs with lymphoma, they increase under malignancy as well as various inflammatory conditions, thereby lowering their diagnostic specificity [17,18]. Other biological enzymes or cytokines released from tumor cells do not have clinical usefulness as diagnostic tools for canine lymphoma [5]. A few commercial biomarkers for detecting lymphoma, including the TK-1 Canine Cancer Panel (VDI Laboratory, Simi Valley, CA, USA) and canine lymphoma blood test (cLBT, Avacta Animal Health, Wetherby, U.K.), are currently available; however, they are not in-house tests and require a few days to yield results [5]. In contrast, the test kit for the ECPKA autoantibody can obtain results relatively quickly with just a drop of serum. Therefore, it is a rapid and useful biomarker for the diagnosis of malignant tumors in clinical practice.

Autoantibody production is said to be induced in response to mutations and degradation, overexpression, and/or release of proteins from damaged cells [19]. Autoantibodies against ECPKA are also thought to be generated by the secretion of PKA C subunits, which originally exist intracellularly. Although the correlation between ECPKA enzymatic activity and the level of ECPKA autoantibody was not documented in this study, there are reports on the application of autoantibodies to measure the activities of biochemical enzymes [15,20,21]. Nesterova et al. developed a novel enzyme immunoassay (EIA) to measure autoantibodies against ECPKA and compared the EIA with the enzymatic assay method [9]. Although they did not demonstrate a correlation between the two assays, the autoantibody titers against ECPKA were higher in cancer patients than in the control group, and autoantibody EIA showed higher sensitivity (90%) and specificity (87%) compared to the enzymatic assay (83% and 80%, respectively) [8]. Thus, the detection of autoantibodies against ECPKA could be used to estimate the level of ECPKA secreted into the extracellular space.

There are some limitations to this study. Firstly, the median age of tumor-bearing dogs was significantly older than that of healthy dogs. Because this was a retrospective study of client-owned dogs, signalment factors such as breed, age, and sex of each group were not thoroughly controlled. However, we showed here that serum levels of ECPKA autoantibodies in dogs with malignant tumors were significantly higher than not only those of healthy dogs, but also those of dogs with benign tumors or non-tumor diseases. This limitation could be addressed by increasing the sample size. Secondly, we could not determine any correlation with serum levels of ECPKA autoantibodies and detailed dogs’ information such as cancer subtypes and affected organs. However, our results showed that the ECPKA could be a universal screening marker for malignant tumors in dogs. In addition, further studies are needed to determine the association between the serum concentration of ECPKA autoantibody and clinical prognosis or outcome in tumor-bearing dogs.

## 5. Conclusions

In summary, serum levels of ECPKA autoantibodies measured by AniScan cancer detection kits from malignant tumor dogs were significantly higher than those from non-malignant tumor, non-tumor, and healthy dogs. The ECPKA autoantibody-based AniScan cancer detection kit could be used as a fairly useful tool for cancer screening in dogs.

## Figures and Tables

**Figure 1 sensors-20-04075-f001:**
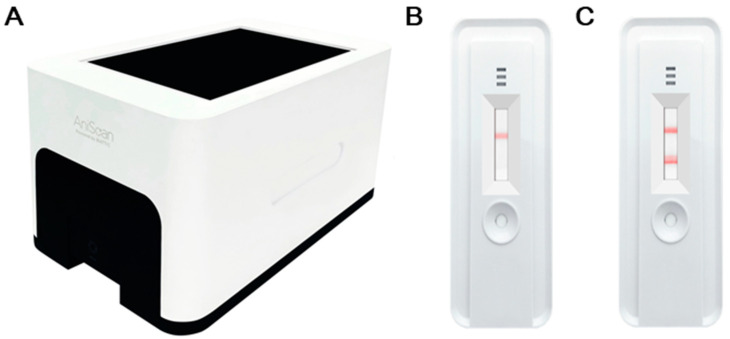
Representative images of the Aniscan cancer detection kit: (**A**) aniscanner, (**B**) a kit from a healthy control sample, and (**C**) a kit from a malignant tumor sample.

**Figure 2 sensors-20-04075-f002:**
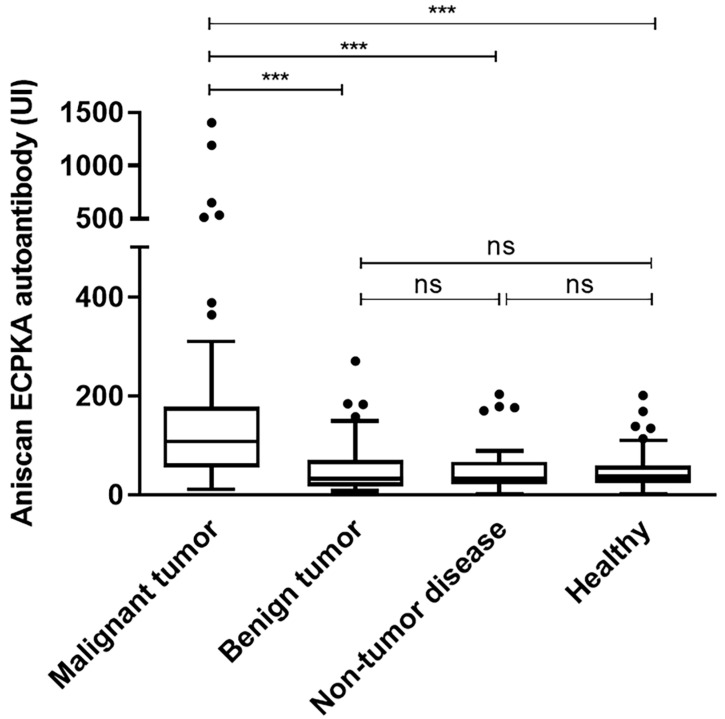
Box plot of AniScan ECPKA autoantibody levels in dogs with malignant tumors, benign tumors, and non-tumor disease and the healthy control group (*** *p* < 0.001 by one-way ANOVA analysis).

**Figure 3 sensors-20-04075-f003:**
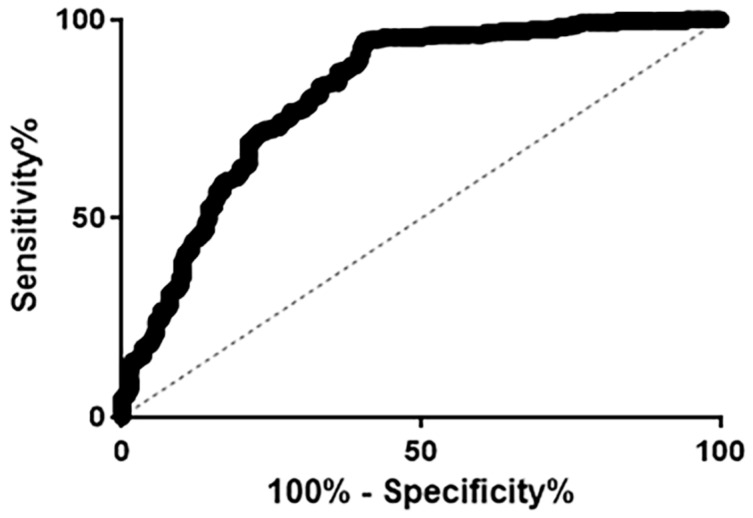
Receiver operating characteristic curve for AniScan ECPKA autoantibody levels.

**Figure 4 sensors-20-04075-f004:**
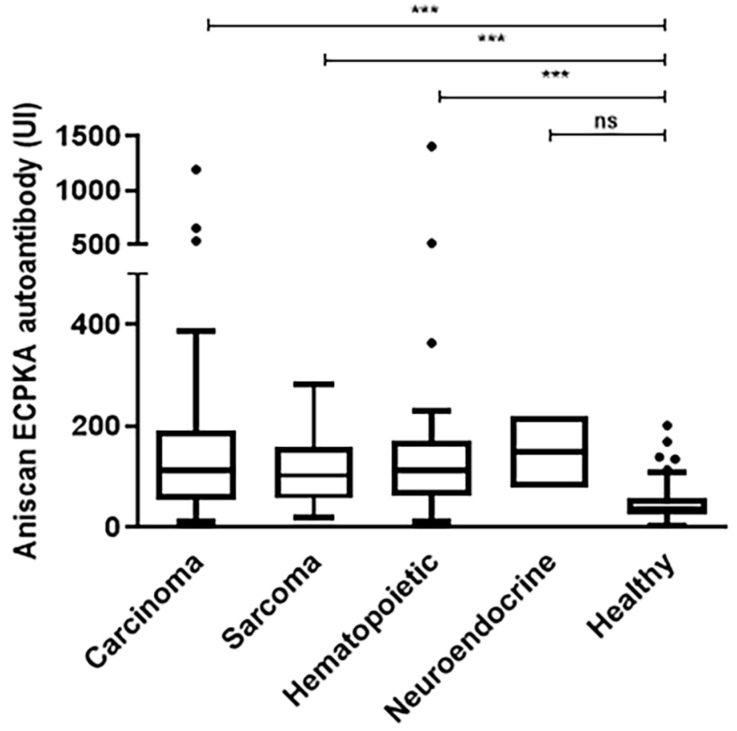
Box and whisker plot of AniScan ECPKA autoantibody levels in dogs with different types of malignant tumors and non-tumor controls (*** *p* < 0.001 by one-way ANOVA analysis).

**Table 1 sensors-20-04075-t001:** Signalment data for dogs enrolled in this study.

	Malignant Tumor	Benign Tumor	Non-Tumor Disease	Healthy
Number	227	60	79	184
Median age (Range)	11 (1–17)	10 (4–17)	8 (1–15)	5 (0.5–13)
Sex (*n*)	F (33), M (16), NF (90), NM (88)	F (7), M (5), NF (13), NM (35)	F (10), M (3), NF (17), NM (49)	F (46), M (42), NF (57), NM (39)
Breed (*n*)	Maltese (48), Shih tzu (25), Yorkshire Terrier (22), Poodle (21), Cocker Spaniel (15), Retriever (13), Schnauzer (11), Pomeranian (10), French Bulldog (3), other (59)	Maltese (10), Shih tzu (8), Cocker Spaniel (7), Beagle (5), Poodle (6), Bichon Frise (2), other (22)	Maltese (20), Yorkshire Terrier (8), Poodle (7), Pomeranian (6), Shih tzu (3), Cocker Spaniel (3), other (32)	Beagle (54), Maltese (29), Shih tzu (26), Pomeranian (11), Yorkshire Terrier (9), Bichon Frise (4), other (51)

Note: F: female, M: male, NF: neutered female, NM: neutered male.

**Table 2 sensors-20-04075-t002:** Disease types and number of dogs with non-tumor disease in this study.

Type of Non-Tumor Disease	Number (*n*)
Orthopedic	19
Dermatologic	17
Gastrointestinal	11
Hepatobiliary	6
Cardiovascular	6
Urologic	5
Endocrine	3
Immune-mediated	3
Ophthalmologic	3
Neurologic	2
Other	4

**Table 3 sensors-20-04075-t003:** The diagnostic abilities of AniScan ECPKA autoantibody levels.

	Value
AUROC	0.818
95% CI	0.777–0.853
Standard error	0.019
Sensitivity	81.08%
Specificity	87.02%
Accuracy	84.67%
PPV	80.35
NPV	87.53

**Note**: AUROC: the area under a receiver operating characteristic curve, CI: confidence interval, PPV: positive predictive value, NPV: negative predictive value.

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
