# Peer review of "AniScan Using Extracellular Cyclic AMP-Dependent Protein Kinase A as a Serum Biomarker Assay for the Diagnosis of Malignant Tumors in Dogs"

_sensors, 2020, doi:10.3390/s20154075_

Round 1

Reviewer 1 Report

Major issues:

In this study the Authors compare the level of autoantibodies against ECPKA in four groups of dogs: with malignant tumors, benign tumors, non-tumor disease and healthy dogs.  The median age of healthy subjects is significantly lower than in the remaining groups. To prove that the level of ECPKA autoantibodies are significantly higher in malignant tumors, it is necessary to demonstrate that there is not a significant correlation between level of autoantibodies and age.

The Authors should consider the comparison between the group of dogs bearing malignant tumors and the healthy dog, avoiding the others (non-tumor disease and benign tumor): the two groups (malignant tumors vs healthy) are more balanced (227 vs 184) and the characteristics of the other groups are not uniform.

The correlation between the levels of ECPKA autoantibodies and certain diseases (line 42-44, page 4, figure 4) is not appropriate considering the small number of patients in each group.

Minor issues:

Table 2: Total number of dogs in non-tumor disease group is 70, while in the text and in table 1 is 79

The maximum range value reported in lines 4 to 6 of page 1 (section of results) is not consistent with the values reported in the figures 1 and 3, and in the text lines 31-33, page 3, where the highest value is more than 1000 UI.

References n° 17 and 18 are not cited properly in the text

Reference n°29 is not cited in the correct order, and there is not a reference n°29 in the references list

The observations reported in lines 99-102, page 6 about the production of antibody and immunoglobulins are not substantiated by solid studies, and maybe they are not necessary to the aim of the discussion.

Author Response

July 11, 2020

 Reviewer #1

Major issues:

  1. COMMENT: In this study the Authors compare the level of autoantibodies against ECPKA in four groups of dogs: with malignant tumors, benign tumors, non-tumor disease and healthy dogs. The median age of healthy subjects is significantly lower than in the remaining groups. To prove that the level of ECPKA autoantibodies are significantly higher in malignant tumors, it is necessary to demonstrate that there is not a significant correlation between level of autoantibodies and age.

RESPONSE: Thank you for raising this important point. It is one of limitations in this study. The median age of with tumor-bearing dogs is significantly older than that of healthy dogs. Because this is a retrospective study of client-owned dogs, signalment factors such as breed, age, and sex of each group are not controlled thoroughly. However, we showed here that serum levels of ECPKA autoantibody of dogs with malignant tumors were significantly higher than not only those of healthy dogs, but also those of dogs with benign tumor or non-tumor diseases. This limitation could be addressed by increasing the sample size. It is described in the revision (discussion section; Page 14, Lane 41). We hope our approach acceptable.

  1. COMMENT: The Authors should consider the comparison between the group of dogs bearing malignant tumors and the healthy dog, avoiding the others (non-tumor disease and benign tumor): the two groups (malignant tumors vs healthy) are more balanced (227 vs 184) and the characteristics of the other groups are not uniform.

RESPONSE: Thank you for raising this important point. The one of aims of this study was to determine whether ECPKA can fulfill the role of malignant tumor detection in dogs. Therefore, we showed here that serum levels of ECPKA autoantibody of dogs with malignant tumors were significantly higher than not only those of healthy dogs, but also those of dogs with benign tumor or non-tumor diseases. We hope our approach acceptable.

  1. COMMENT: The correlation between the levels of ECPKA autoantibodies and certain diseases (line 42-44, page 4, figure 4) is not appropriate considering the small number of patients in each group.

RESPONSE: Thank you for your detailed comment, and we agree with you. Further studies with large sample size are needed to discuss this issue. We removed the text and figure in the revision.

Minor issues:

  1. COMMENT: Table 2: Total number of dogs in non-tumor disease group is 70, while in the text and in table 1 is 79

RESPONSE: Thank you for your detailed comment. It was our mistake, and the table 2 was revised in the revision.

  1. COMMENT: The maximum range value reported in lines 4 to 6 of page 1 (section of results) is not consistent with the values reported in the figures 1 and 3, and in the text lines 31-33, page 3, where the highest value is more than 1000 UI.

RESPONSE: Thank you for your detailed comment. It was our mistake. The median value in the malignant tumor group was 135.12 (range: 11.03–1406.10) UI, and in the benign tumor, non-tumor disease, and healthy control groups, the values were 49.88 (range: 8.10–270.60) UI, 45.98 (range: 1.00–203.60) UI, and 44.23 (range: 1.70–200.80) UI, respectively. It has been described in the revision (Results section; Page 7, Lane 3).

  1. COMMENT: References n° 17 and 18 are not cited properly in the text

RESPONSE: Thank you for your detailed comment, and we revised text and reference section in the revision.

  1. COMMENT: Reference n°29 is not cited in the correct order, and there is not a reference n°29 in the references list

RESPONSE: Thank you for your detailed comment, and we revised text and reference section in the revision.

  1. COMMENT: The observations reported in lines 99-102, page 6 about the production of antibody and immunoglobulins are not substantiated by solid studies, and maybe they are not necessary to the aim of the discussion.

RESPONSE: Thank you for your suggestion, and we agree with you. We removed the paragraph in the revision.

Again, we appreciate all of your insightful comments. We worked hard to respond to them. Thank you for taking the time and energy to help us improve this manuscript.

Sincerely yours,

Ul-Soo Choi, DVM, PhD

Department of Veterinary Clinical Pathology and Bio-Safety Research Institute, College of Veterinary Medicine, Chonbuk National University, Jeonju, Republic of Korea

E-mail: uschoi@jbnu.ac.kr

Dong-Ha Bhang, DVM, PhD

Department of Molecular and Cellular Biology, Samsung Biomedical Research Institute, Sungkyunkwan University School of Medicine, Suwon, Republic of Korea

E-mail: bhangd77@gamil.com

Reviewer 2 Report

This is an excellent manuscript demonstrating the use of the novel AniScan detection kit for the diagnosis of cancer in dogs. The analysis is based on the serum detection of extracellular c-AMP-dependent protein kinase A (ECPKA) autoantibodies by lateral flow immunochromatography making it a rapid and convenient approach to reveal both solid and blood canine tumors. The manuscript is very accurately written and the data are sufficient to support the author's conclusions. There are only minor deficits present that can be easily dealt with by the authors.

1) It is not clear what does "§" designate in the names of the first two authors Ji-Eun Lee and Woo-Jin Song?

2) It is not clear what is the colloidal gold conjugate conjugated to (line 104)?

3) Would it be possible to disclose to the readers in the Methods section the sequence of the ECPKA antigen used (line 105)? If not, could the reason why this is not possible be stated in the text?

4) Similarly, could the authors be more specific regarding the ECPKA antigen and the anti-canine IgG (line 105)? It would be particularly useful to state in the text how or from where these components were obtained.

5) In addition, it would be helpful to specify exactly how intensities of the resulting red lines are being collected (line 106)?

6) Would it be possible to provide a representative image of how the readout from the AniScan cancer detection kit looks like between malignant tumor and healthy samples as part of a new figure?

7) Please explain based on what statitistical criteria were the box and whisker plots of Figures 1, 3, and 4 constructed in the Methods section/Statistical analysis or as part of the respective figure legends.

8) Moreover, it is not clear whether median is being plotted in Figures 1, 3, and 4 as the box plot values do not correspond to the median values listed in the text (lines 3–6 for Figure 1 and lines 30–34 for Figure 3). Please revise.

9) It is not clear from the introductory Results section (page 3) what particular organs were affected by the given solid cancers in dogs? Could the authors please corroborate on that?

10) The first column of Table 1 needs reformatting as it is too narrow and thereby difficult to read.

11) There is a conflict between the following sentence (line 28) and the data presented in Figure 3: "All types of malignant tumors, carcinomas, sarcomas, hematopoietic cancers, and neuroendocrine cancers showed significant differences in the level of ECPKA autoantibodies when compared to non-tumor controls (all, p < 0.01) (Figure 3)". Whereas the sentence claims that neuroendocrine cancers showed significant difference, these are labeled as non-significant in Figure 3. Please reconcile this discrepancy.

12) Would the AniScan ECPKA autoantibody level test be still predictive if samples were stratified based on the following criteria?

a) different cancer sub-types

b) different organ affected in solid cancers

c) different dog breeds

Examples of such stratification could be presented as part of new figure(s) introduced into the manuscript.

13) Please give examples in the respective figure legend to what non-tumor diseases does "Etc." in Figure 4 refer to.

14) Could the authors briefly comment in the Discussion section on whether increased serum ECPKA autoantibody was also associated with poor survival prognosis or therapy outcome?

15) Please define the following abbreviations: "SAA" (line 66), "CRP" (line 66), and "CHOP" (line 96).

16) The concluding sentence "In summary, barring some limitations in using autoantibodies, ECPKA autoantibody level is a fairly useful biomarker, with good diagnostic parameters, for detecting malignant tumors in dogs." (line 107) is not precise in that it does not focus on the novel discovery of the present work, which is that AniScan cancer detection kit can detect ECPKA autoantibodies as a cancer diagnostic. The fact that ECPKA is a biomarker for malignant tumors in dogs was already found by the paper listed as reference 13. Similarly the sentence "This is consistent with the findings of the present study; thus, ECPKA can be used as a universal cancer biomarker and applied to screening malignant tumors from benign tumors or other diseases." (line 57) should be rephrased in a manner acknowledging the previous study and including a citation to reference 13.

Author Response

July 11, 2020

Reviewer #2

This is an excellent manuscript demonstrating the use of the novel AniScan detection kit for the diagnosis of cancer in dogs. The analysis is based on the serum detection of extracellular c-AMP-dependent protein kinase A (ECPKA) autoantibodies by lateral flow immunochromatography making it a rapid and convenient approach to reveal both solid and blood canine tumors. The manuscript is very accurately written and the data are sufficient to support the author's conclusions. There are only minor deficits present that can be easily dealt with by the authors.

  1. COMMENT: It is not clear what does "§" designate in the names of the first two authors Ji-Eun Lee and Woo-Jin Song?

RESPONSE: Thank you for your detailed comment. We described in the revision: §These first two authors contributed equally to this work.

  1. COMMENT: It is not clear what is the colloidal gold conjugate conjugated to (line 104)?

RESPONSE: Thank you for your detailed comment. Serum ECPKA autoantibodies conjugated with 50 nM gold nanoparticles were bound by canine PKA antigen pre-coated on the test line. Then, the result appeared as red lines. It has been described in the revision (Materials and Methods section; Page 3, Lane 22).

  1. COMMENT: Would it be possible to disclose to the readers in the Methods section the sequence of the ECPKA antigen used (line 105)? If not, could the reason why this is not possible be stated in the text?

RESPONSE: Thank you for your detailed comment. We used the sequence, National Center for Biotechnology Information reference sequence: NM_001003032.1, for the canine PKA antigen. It has described in the revision (Materials and Methods section; Page 3, Lane 23).

  1. COMMENT: Similarly, could the authors be more specific regarding the ECPKA antigen and the anti-canine IgG (line 105)? It would be particularly useful to state in the text how or from where these components were obtained.

RESPONSE: Thank you for your detailed comment. We used the canine PKA antigen sequence as described above. And, it was our mistake that we have mentioned the anti-canine IgG which was not used in this study. It has revised in the revision (Materials and Methods section; Page 3, Lane 23). We hope our approach acceptable.

  1. COMMENT: In addition, it would be helpful to specify exactly how intensities of the resulting red lines are being collected (line 106)?

RESPONSE: Thank you for your suggestion. The result appeared as red lines, and strength of red color was measured by using computer vision techniques (Aniscaner, Biattic Inc.) that could find the exact location of red lines on the kit image. The system calculates the areas of the red lines, and the values of the areas stand for intensities of the lines. It has been described in the revision (Figure 1; Materials and Methods section; Page 3, Lane 24).

  1. COMMENT: Would it be possible to provide a representative image of how the readout from the AniScan cancer detection kit looks like between malignant tumor and healthy samples as part of a new figure?

RESPONSE: Thank you for your suggestion, and we agree with you. We added a new figure in the revision (Figure 1).

  1. COMMENT: Please explain based on what statitistical criteria were the box and whisker plots of Figures 1, 3, and 4 constructed in the Methods section/Statistical analysis or as part of the respective figure legends.

RESPONSE: Thank you for your detailed comment. All graphs are presented as box and whisker plots. All the data are shown as the median and range obtained in at least three independent experiments. The statistical analyses were performed using SPSS Statistics 25 (IBM Corp., Chicago, IL, USA) and GraphPad Prism version 8 (GraphPad Software, La Jolla, CA, USA). A p-value of 0.05 was considered statistically significant. It has been described in the revision (Materials and Methods section; Page 5, Lane 4). We hope our approach acceptable.

  1. COMMENT: Moreover, it is not clear whether median is being plotted in Figures 1, 3, and 4 as the box plot values do not correspond to the median values listed in the text (lines 3–6 for Figure 1 and lines 30–34 for Figure 3). Please revise.

RESPONSE: Thank you for your detailed comment, and it was our mistake. We have revised it in the revision.

  1. COMMENT: It is not clear from the introductory Results section (page 3) what particular organs were affected by the given solid cancers in dogs? Could the authors please corroborate on that?

RESPONSE: Thank you for raising this important comment. It is one of limitations in our study. We could not determine correlation with serum levels of ECPKA autoantibody and detailed dogs’ information such as cancer subtypes and affected organs. However, our results showed that the ECPKA could be a universal screening marker for malignant tumor in dogs. It has been described in the revision (Discussion section; Page 15, Lane 2). We hope our approach acceptable.

  1. COMMENT: The first column of Table 1 needs reformatting as it is too narrow and thereby difficult to read.

RESPONE: Thank you for your detailed comment, and we revised the Table 1 as your comment.

  1. COMMENT: There is a conflict between the following sentence (line 28) and the data presented in Figure 3: "All types of malignant tumors, carcinomas, sarcomas, hematopoietic cancers, and neuroendocrine cancers showed significant differences in the level of ECPKA autoantibodies when compared to non-tumor controls (all, p < 0.01) (Figure 3)". Whereas the sentence claims that neuroendocrine cancers showed significant difference, these are labeled as non-significant in Figure 3. Please reconcile this discrepancy.

RESPONSE: Thank you for raising this important point. All types of malignant tumors except neuroendocrine cancers showed significant differences in the level of ECPKA autoantibodies when compared to non-tumor controls (all, p < 0.001). It has been revised in the revision (Results section; Page 12, Lane 1).

  1. COMMENT: Would the AniScan ECPKA autoantibody level test be still predictive if samples were stratified based on the following criteria?
  2. a) different cancer sub-types
  3. b) different organ affected in solid cancers
  4. c) different dog breeds

RESPONSE: Thank you for raising this important point. There are some limitations in this study. First, the median age of with tumor-bearing dogs is significantly older than that of healthy dogs. Because this is a retrospective study of client-owned dogs, signalment factors such as breed, age, and sex of each group are not controlled thoroughly. However, we showed here that serum levels of ECPKA autoantibody of dogs with malignant tumors were significantly higher than not only those of healthy dogs, but also those of dogs with benign tumor or non-tumor diseases. This limitation could be addressed by increasing the sample size. Second, we could not determine correlation with serum levels of ECPKA autoantibody and detailed dogs’ information such as cancer subtypes and affected organs. However, our results showed that the ECPKA could be a universal screening marker for malignant tumor in dogs. It has been described in the revision (Discussion section; Page 14, Lane 41). We hope our approach acceptable.

Examples of such stratification could be presented as part of new figure(s) introduced into the manuscript.

  1. COMMENT: Please give examples in the respective figure legend to what non-tumor diseases does "Etc." in Figure 4 refer to.

RESPONSE: Thank you for your detailed comment. As another reviewer’s comment, we removed the text and figure of non-tumor disease group. We agree to the reviewer, and further studies with large sample size are needed to discuss this issue. We hope our approach acceptable.

  1. COMMENT: Could the authors briefly comment in the Discussion section on whether increased serum ECPKA autoantibody was also associated with poor survival prognosis or therapy outcome?

RESPONSE: Thank you for your great suggestion. Unfortunately, we did not evaluate correlations with serum levels of ECPKA autoantibody and clinical prognosis or outcome in tumor-bearing dogs in this study. Further studies should be needed to discuss this issue. It has been described in the revision (Discussion section; Page 15, Lane 5). We hope our approach acceptable.

  1. COMMENT: Please define the following abbreviations: "SAA" (line 66), "CRP" (line 66), and "CHOP" (line 96).

RESPONSE: Thank you for your detailed comment. We have revised it in the revision.

  1. COMMENT: The concluding sentence "In summary, barring some limitations in using autoantibodies, ECPKA autoantibody level is a fairly useful biomarker, with good diagnostic parameters, for detecting malignant tumors in dogs." (line 107) is not precise in that it does not focus on the novel discovery of the present work, which is that AniScan cancer detection kit can detect ECPKA autoantibodies as a cancer diagnostic. The fact that ECPKA is a biomarker for malignant tumors in dogs was already found by the paper listed as reference 13. Similarly the sentence "This is consistent with the findings of the present study; thus, ECPKA can be used as a universal cancer biomarker and applied to screening malignant tumors from benign tumors or other diseases." (line 57) should be rephrased in a manner acknowledging the previous study and including a citation to reference 13.

RESPONSE: Thank you for your suggestion, and we agree with you. Our results have revealed that’ serum levels of ECPKA autoantibody measured by Aniscan cancer detection kit from malignant tumor dogs were significantly higher than those from non-malignant tumor, non-tumor, and healthy dogs. Therefore, the ECPKA autoantibody-based Aniscan cancer detection kit could be used as a fairly useful tool for cancer screening in dogs. It has been described in the revision (Conclusion section; Page 15, Lane 9). We hope our approach acceptable.

Again, we appreciate all of your insightful comments. We worked hard to respond to them. Thank you for taking the time and energy to help us improve this manuscript.

Sincerely yours,

Ul-Soo Choi, DVM, PhD

Department of Veterinary Clinical Pathology and Bio-Safety Research Institute, College of Veterinary Medicine, Chonbuk National University, Jeonju, Republic of Korea

E-mail: uschoi@jbnu.ac.kr

Dong-Ha Bhang, DVM, PhD

Department of Molecular and Cellular Biology, Samsung Biomedical Research Institute, Sungkyunkwan University School of Medicine, Suwon, Republic of Korea

E-mail: bhangd77@gamil.com

Round 2

Reviewer 1 Report

The Authors fulfilled all the requests for major and minor revisions. As regards the point N°1 - major issues, the Authors did not answer completely, since they did not prove that there is not correlation between age and the level of ECPKA autoantibodies.

However, the comments in the revised version can be considered acceptable, as the Authors highlighted the limitation of the study and they confirmed that it is necessary a larger sample size and a prospective study.